# Mental health of Urban Mothers (MUM) study: a multicentre randomised controlled trial, study protocol

Simone Eliane Schwank ,[1,2] Ho-Fung Chung,[3] Mandy Hsu,[4] Shih-Chien Fu,[5] Li Du,[6] Liping Zhu,[6] Hsuan-Ying Huang,[7] Ewa Andersson,[8] Ganesh Acharya[2,9]

For numbered affiliations see end of article.

**Correspondence to**
Dr Simone Eliane Schwank, CLINTEC, Karolinska Institute, Stockholm, Sweden; simone.schwank@ki.se

## ABSTRACT

**Introduction** Mental health disorders are common during pregnancy and the postnatal period and can have serious adverse effects on women and their children. The consequences for global mental health due to COVID-19 are likely to be significant and may have a long-term impact on the global burden of disease. Besides physical vulnerability, pregnant women are at increased risk of mental health problems such as anxiety, depression and post-traumatic stress disorder due to the consequences of social distancing. It can result in altered healthcare routines, less support from the family and friends, and in some cases, partners not being allowed to be present during prenatal visits, labour and delivery. Higher than expected, rates of perinatal anxiety and depression have been already reported during the pandemic. Pregnant women may also feel insecure and worried about the effects of COVID-19 on their unborn child if they get infected during pregnancy. Today, young urban women are used to using internet services frequently and efficiently. Therefore, providing mental health support to pregnant women via internet may be effective in ameliorating their anxiety/depression, reducing the risk of serious mental health disorders, and lead to improved maternal and perinatal outcomes.

**Overarching aim** Our aim is to explore the effectiveness of a web-based psychosocial peer-to-peer support intervention in reducing the risk and severity of perinatal mental health disorders and preventing adverse pregnancy outcomes among pregnant women living in metropolitan urban settings.

**Methods and analysis** We plan to conduct a multicentre prospective randomised controlled trial, Mental health of Urban Mothers trial. Pregnant women living in large metropolitan cities will be recruited using internet-based application through non-profit organisations' websites. The women who consent will be randomised to receive a web-based peer-to-peer support intervention or usual care. Data will be analysed to identify the effects of intervention on Edinburgh Postnatal Depression Score and Generalised Anxiety Disorder 7 scores as well as pregnancy outcomes. The impact of COVID-19 pandemic on maternal stress will be assesed using Impact Event Scale-R. Any differences in outcomes between cities will be addressed in subgroup analyses.

**Ethics and dissemination** The study will be conducted according to the principles of Good Clinical Practice and will follow the ethical principles of the Declaration

## Strengths and limitations of this study

► A evidence-based 'Thinking Healthy' intervention will be applied using state-of-the-art eHealth peer-to-peer support to elucidate the role of internet-based virtual consultation and psychosocial support on maternal mental health.
► Our multiprofessional research team will ensure ethical standards along with the quality of intervention, data collection and follow-up.
► An external data monitoring committee will monitor the trial and data quality.
► Due to the nature of intervention, investigators, as well as the study participants, will not be blinded to the intervention.
► Different cultures and healthcare systems in different study locations may have different impacts on pregnancy outcomes.

of Helsinki. The study protocol has been approved by the ethical review board of Chinese University of Hong Kong (IRB number 2019-8170) and Shanghai Center for Women's and Children's Health (international review board (IRB) number 2020-F001-12). The results will be disseminated at national and international scientific conferences, published in peer-reviewed medical journals and spread to the public through social media, news outlets and podcasts.

**Trial registration number** NCT04363177; Trial sponsor Karolinska Institute, CLINTEC, Stockholm, Sweden.

## INTRODUCTION

Pregnancy is a period of transition and great change, which may make some women more vulnerable to mental health problems. It is known that depressive symptoms during pregnancy may influence birth outcomes.[1–3] The prevalence of postpartum depression (PPD) is increasing globally but varies considerably between different countries and cultures.[4 5] In China, the prevalence of PPD in women of the one-child generation is reported to be higher compared with previous generations, particularly among women giving birth to a daughter.[6] Prevalence rates are highest in

the rural areas' prefecture-level cities (25.4%), lower in the provincial capitals (19.5%) and lowest in the municipalities directly under central government (12.9%).[7 8] Furthermore, there are substantial gaps in the accessibility of care.[2] Perinatal mental health disorders may become more prevalent during a time of acute crisis, and the prevalence of maternal anxiety, distress and PPD can be expected to increase in China and elsewhere as a result of the COVID-19 pandemic.[9 10] In this context, a recent study found that 35.4% of the participating women had an Edinburgh Postnatal Depression score (EPDS) >13.[11] However, there might be cross-national differences in the risk factors and impact of pandemic on the prevalence of perinatal mental health disorders.

Some pregnant women might be predisposed to post-traumatic stress disorder (PTSD) during a crisis situation, such as the COVID-19 pandemic. Mothers who developed PTSD in response to the 9/11 terrorist attacks had lower morning and evening salivary cortisol levels, compared with mothers who did not develop PTSD.[12] Beyond effects on the mother alone, perinatal mental health issues can have long-term effects on child's mental and physical health as well as their behaviour and cognition.[13–17] Distress in pregnant women may affect the foetus and is known to induce epigenetic changes in the placental genes.[18] Increased risk of psychopathology is observed in children exposed to maternal prenatal distress. Elevated maternal cortisol and epigenetic regulation of placental glucocorticoid-pathway genes are potential mechanisms for these observations.[19] Women often express feelings of inadequacy in the new mothering role, which can furthermore negatively impact their mental health and relationship to their infant.[20]

Effective treatments are needed to address high global rates of PPD with onset typically within 4 weeks after delivery[21] and maternal depression up to 2 years after delivery.[22 23] Programmes aimed at reducing PPD could achieve the most cost-efficient results by focusing efforts in the critical time periods around childbirth.[22] Web-based psychosocial support provided by trained public health nurses is an effective treatment method for PPD.[24] Limited public health resources are challenges to the accessibility of mental health services, which is why the use of web-based psychosocial support could be a good alternative. Women perceive the risk for themselves or their infants to be above average during global crises, which increases the levels of uncertainty. However, face-to-face consultations during a pandemic are likely to increase the risk of disease transmission. Therefore, easily accessible eHealth support could provide fast and resource-effective care during the COVID-19 pandemic.

## Previous research

Screening programmes appear to be effective in identifying pregnant women at risk of developing perinatal mental health disorders,[25] but they are neither universally implemented in public healthcare nor necessarily lead to effective interventions. One reason for the limited

effectiveness is the small number of multidisciplinary perinatal mental healthcare teams that provide for the assessment and treatment of perinatal mental health disorders.[26] Pregnant women in Shanghai reported a lack of access to resources and a strong preference for web-based psychosocial support. Previous studies suggested that women who received a peer support intervention were at half the risk of developing postnatal depression at 12 weeks postpartum than those in a control group.[27 28] A recent systematic review has shown that therapist-supported internet-based cognitive behavaroral therapy (iCBT) significantly improves stress, anxiety and depressive symptoms among postpartum women, although the magnitude of effect was variable.[29]

There are challenges associated with supporting a family and raising children for young people living in modern metropolitan urban areas. These challenges may have a negative effect on maternal mental health. In Shanghai, China, a previous Randomized control trial (RCT) at the Obstetrics and Gynaecology Hospital of Fudan University concluded that a prenatal evidence-based psychosocial intervention model for high-risk pregnant women holds potential as a preventive programme that can address maternal health and improve birth outcomes.[30] The mental health statuses of 6024 pregnant women were investigated when they registered for prenatal care from September 2015 to August 2016, where 23.89% and 9.44% had symptoms of depression and anxiety, respectively.[30 31] There is a significant positive association between parity, depression, older age and risk of developing PPD.[32 33] A cross-sectional study of 842 pregnant women with obstetric complications attending the same hospital reported that the prevalence of major and minor depression in high-risk pregnant women during antenatal period was 8.3% and 28.9%, respectively.[32]

Young couples in Hong Kong appear to have difficulties in balancing their traditional family roles and expectations with the hardships of living in a modern society. A recent study has found high levels of perinatal anxiety in families due to the financial burden caused by having children to honouring traditional family roles. Having children is highly encouraged and a sign of prosperity in a traditional Chinese family, yet, it is very expensive to raise even one child in Hong Kong.[34] Having a high level of work–family conflict is a strong predictor of mental health problems across pregnancy. Family–work balance was also found to be notoriously poor in Hong Kong families, due to the extremely high demands of working hours. Dual-earner families in Hong Kong indicated that the work–family conflict was negatively related to job and life satisfaction. The study showed that the coping behaviours of Hong Kong employed parents were largely ineffective in solving problems related to work–family conflicts.[35]

## OBJECTIVES

### Purpose and aims

Our aim is to explore the effectiveness of a web-based psychosocial peer-to-peer support intervention in reducing the risk and severity of perinatal mental health disorders and preventing adverse pregnancy outcomes among pregnant women residing in metropolitan urban settings.

### Primary objective

To investigate the impact of a web-based psychosocial intervention on EPDS among urban pregnant women living in Hong Kong and Shanghai.

### Secondary objectives

1. To investigate the impact of a web-based psychosocial intervention on maternal anxiety levels assessed by Generalised Anxiety Disorder 7 (GAD7) scores.
2. To explore the impact of COVID-19 pandemic on maternal stress levels assessed by Impact Event Scale (IES-R).
3. To explore the impact of web-based psychosocial intervention on mental health and pregnancy outcomes among women from different socioeconomic and cultural environments.
4. To explore the relationship between mental health status of pregnant women and rates of elective caesarean section on maternal request.

## MATERIAL AND METHODS

### Trial design and setting

The Mental health of Urban Mothers trial is a prospective randomised controlled trial that will be performed in order to assess the effects of psychosocial support during pregnancy that focuses on factors that can reduce mental health problems and adverse outcomes in pregnant and postpartum women (intervention) versus standard care (control group). The intervention will be delivered by trained peers. Participating women will be recruited from two large urban areas, that is, Hong Kong (7.5 million inhabitants and among all maternity hospitals 11 027 annual deliveries[36] and Shanghai (25 million inhabitants and Shanghai Maternity and Child Healthcare Hospital with more than117 700 deliveries per year). The intervention group will receive web-based psychosocial peer-to-peer support two times during the antenatal period before 36 gestational weeks (at 18–22 weeks and 24–28 weeks) and the control group will receive the usual standard antenatal care. Mental health status will be assessed using standard questionnaires, that is, EPDS and GAD7 and IES-R. Information on the course and outcome of pregnancy will be obtained by sending a questionnaire at 4–6 weeks postnatally via a web survey.

### Hypothesis

Feelings of acute stress, anxiety, depression and more severe negative emotions such as PTSD that may lead to adverse pregnancy outcomes can be ameliorated by an appropriate internet-based psychosocial intervention started early in pregnancy, which will in turn influence the women's overall mental well-being and improve pregnancy outcomes.

### Recruitment and blinding

Pregnant women will be recruited via non-profit organisation websites in each country. In addition to Hong Kong and Shanghai, we aim to include additional urban locations, including Stockholm, Sweden in this multicenter study. Via web link, the women will be guided to the research project website. The peer support website includes information about the project and informed consent. By completing the initial sociodemographic survey, the women can consent to participate in the research study. Then, they will ask to use an app to follow the study and complete the survey questionnaires. The participating women will be automatically randomised to intervention (peer-to-peer web-based psychosocial support) or control group (standard care) by the computer-generated random sequence. Regardless of recruitment to the intervention or control group, the women will continue their regular visits at the maternity clinics as usual.

Both the women and their peers will be forbidden to inform anyone in the study about which study group they belonged to. However, due to the nature of intervention, investigators, as well as the study participants, will not be blinded to the intervention.

### Inclusion criteria

Pregnant women between 12 and 18 weeks of gestation with viable intrauterine pregnancy.

### Exclusion criteria

History of severe psychiatric or substance abuse disorder requiring medical treatment or presence of fetal chromosomal/structural abnormality.

### Recruitment pathway

The participant recruitment pathway and the follow-up during pregnancy are described in figure 1.

### Intervention

The intervention is a web-based psychosocial peer-to-peer support during pregnancy. It will be administered using WhatsApp and WeChat APP to contact pregnant women and provide psychosocial support over chat, call and video function.

The evidence-based eHealth peer-to-peer psychosocial intervention 'Thinking Healthy' will be tested in this RCT. In line with the WHO's mental health gap (mhGAP) Intervention Guide, 'Thinking Healthy' is used to identify and manage perinatal mental health conditions (distress, symptoms of depression and anxiety) in non-specialised psychosocial support setting.[37] 'Thinking Healthy' includes guidance on evidence-based interventions to identify and manage a number of priority mental health conditions. 'Thinking Healthy' has been widely

| Year | Jan-March | April-June | July-September | October-November | November-December |
|------|-----------|------------|----------------|------------------|-------------------|
| 2020 | Protocol for survey study<br><br>IRB Hong Kong, Shanghai,<br><br>Set up Survey | **T1: First peer-to-peer intervention.** Participants recruitment. (200 participants for each city)<br>Randomization (antenatal < week 18).<br>Web-based Peer-to-peer support at 18-22 weeks | **T2: Second peer-to-peer intervention.**<br>Web-based Peer-to-peer support at 24-28 weeks | **T3: post intervention** (34-36 weeks of gestation). Psychological tests: EPDS, GAD7, IES-R, COVID-19 19 survey | **T4: post intervention** (postpartum approx. 4-6 weeks after childbirth). Psychological tests: EPDS, GAD7, IES-R, COVID-19 survey |
| 2021 | Data collection completed | Analysis and interpretation of data | Report/manuscript writing | Reports/manuscript writing | Submission/Publication of results |

**Figure 1** Overview of the study and timeline. EPDS,Edinburgh Postnatal Depression Score; GAD7, Generalised AnxietyDisorder 7; IES-R, Impact Event Scale-R.

used in low and low-middle income countries with limited mental health professionals.[38] One of such priority conditions is perinatal depression. 'Thinking Healthy'" also uses strategies (plans and activities) taken from iCBT[39] to bring about a change in the mothers' symptoms and functioning. The training does not turn the non-specialised health workers or trained peers into cognitive behavioural therapists—it only allows them to apply these strategies in their day-to-day work with women in the perinatal period. Non-specialised health workers or trained peers can practice 'Thinking Healthy'.[40] The research team provides support for the non-specialised healthcare workers and peers. In our RCT, the web-based psychological support will be provided by trained peers using diverse web-based tools, including video calls, chat messaging and voice calls for both women and their partners.

### Intervention group
They will receive a web-based psychosocial peer-to-peer support, using 'Thinking Healthy' two times during pregnancy (see figure 1).

### Control group
They will receive standard perinatal care provided at maternity clinics.

### Outcome measures
The primary outcome measure is the woman's EPDS assessed at 34–36 weeks of gestation and at 4–6 weeks postnatally.

Secondary outcomes are GAD7 scores to assess maternal anxiety levels, IES-R score to assess PTSD and pregnancy complications, including occurrence of mental health disorders, rate of preterm birth, rate of induction of labour, rate of vaginal birth, elective caesarean sections on maternal request as well as other birth outcomes.

### Psychological assessments
EPDS[41 42] is a widely validated screening instrument. A high EPDS score does not in itself confirm a diagnosis of depression. A score above a certain cut-off point may, however, indicate probable depressive disorder. In the present study, the EPDS will be administered in the third trimester of pregnancy (a 34–36 weeks) and at 4–6 weeks postpartum. An EPDS cut-off score of >10 has been found to be optimal for perinatal screening for depression.[43]

*The* GAD7 questionnaire[44 45] has been translated into Chinese and validated in general hospital outpatients, with a cut-off score of >10[22] to indicate anxiety.

IES-R will be used to assess PTSD. It has been translated and validated in Chinese[46–48] to assess symptoms of PTSD corresponding with Diagnostic and Statistical Manual of Mental Disorders (DSM-V).[46 47] A survey about COVID-19 will also be administered at the same time points (at 34–36 weeks of gestation and 4–6 weeks postpartum).

The EPDS, GAD7 and IES-R scales were chosen over psychodiagnositc interviews because they show high reliability and validity and are more cost-effective and efficient compared with psychodiagnostic interviews.[49]

### Sample size and power calculation
The sample size was calculated using an online sample size calculator (CinCalc.com) to detect a 20% difference in mean EPDS scores between the intervention versus control groups with a 80% power and an alpha of 0.05 in each study location.[50] EPDS scores show a normal distribution with a mean of 12.5 and SD of 4.0.[51] A difference of 20% in the primary outcome is of clinical relevance with regards to the symptomatology of PPD.[52] Based on this, the calculated sample size is 40 participants in each arm (intervention/control) of the study in each study location (n=80 participants per city). Taking into account and possible dropouts, we estimated a required sample size of 100 participants for each city (study location). The same principle will be applied if any additional study locations are added to this multicentre RCT.

## Data management

Each study participant will receive a study number for identification. Data will be collected continuously and through the smartphone application entered into an electronic case report system. Data will be encrypted and safely stored on an electronic server at the Karolinska Institute in Stockholm, Sweden in compliance with European general data protection regulation. Anonymised data will be extracted from the electronic case report system after data collection has ended and analysed for the different endpoints.

## Statistical methods

Data will be analysed per protocol using an intention-to-treat method. Average EPDS, GAD7 and IES-R scores will be compared between the intervention and control groups and across time points (prenatal and postnatal), using independent sample t-tests. Differences in proportions will be assessed using $\chi^2$ test. The non-normally distributed continuous variables will be analysed with the Mann-Whitney U test, and the results of the intervention will be analysed by Cohen's effect size. Multiple imputations will be used for missing data. The variables used in the imputation mode will be previous EPDS score, education level and parity. All patients' demographic data, retrieved from web survey, will be analysed by descriptive (mean, median and range) statistics. For categorical variables, absolute frequency, percentages and/or proportions will be calculated. $\chi^2$ or Fisher's exact test will be used for comparison between groups. For the primary outcome measure, a two-sided p value of <0.05 will be considered significant. Subgroup analyses will be performed for several subgroups of EPDS scores. Spearman's correlation test will be used to test the correlation between maternal antenatal mental health characteristics assessed by EPDS, GAD7 and IES-R at <34–36 weeks and at 6 weeks postpartum.

### Analysis plan

The EPDS, GAD7 and IES-R scores between intervention and control groups will be directly compared. We will also compare the proportions of women with EPDS score of >10 and>15 in the two groups.

### Confounders

Socioeconomic status, educational level, parity, obstetric history including history of complications and trauma during previous pregnancies, number of surviving children, history of chronic diseases and use of medication, complications developing during current pregnancy (such as preeclampsia, gestational diabetes etc) may all be confounders.

### Patient and public involvement

No patients or public have been involved in the design, recruitment or conduction of the study.

### Significance

Our study will generate evidence on whether web-based early intervention programmes could be efficient in ameliorating the risk and severity of perinatal mental health disorders and inform best clinical practice for women affected by the consequences of the COVID-19 pandemic.

This multicentre RCT to be conducted in Hong Kong and Shanghai is expected to provide access to an adequate sample size and external validity as well as insight into socioeconomic and cultural specifics of perinatal mental health disorders. Mental health disorders contribute up to 30% of the global burden of disease[53] and their prevalence is expected by rise during the COVID-19 pandemic. Intergenerational transmission of maternal perinatal mental health disorders has been shown to impact the child's physical and mental development.[19 54] These risks might be heightened in the time of the COVID-19 pandemic. Our RCT intends to provide some insight into consequences of the COVID-19 pandemic on pregnant women's mental health status and pregnancy outcomes.

This interdisciplinary collaboration project may fill the research gap regarding the relationship between mental health and social environments. The study will raise awareness of mental healthcare and highlight the feasibility and value of web-based psychosocial support as a cost-effective way of preventing/ameliorating mental health disorders in mothers and their families during the COVID-19 pandemic.

### Equipment and infrastructure

Hong Kong's maternity hospitals and the Shanghai Maternity and Child Healthcare Hospitals have about 11 027 and 117 700 respective deliveries annually. All the places have the necessary infrastructure to facilitate patient recruitment for research, conducting interviews and distributing survey questions.

## ETHICS AND DISSEMINATION
### Adverse events

Peer to peer support is widely used for mental health promotion among adolocent and women. Previous research has demonstrated that peer supportive relationships help contribute to positive adjustment and to buffer against stressors and adversities, including psychiatric problems.[28]

No serious adverse effects are expected to occur as a result of our intervention, that is, web-based psychological support. Any adverse event, defined as a serious undesirable experience/event or injury associated with the use of the study intervention, will be reported to the health authorities of the respective study location and principal investigator. All participants will be covered by the national health insurance plans of the respective study sites.

### Monitoring

The data monitoring committee for the RCT consists of Professor Xu Biao, Karolinska Institutet, Stockholm,

Sweden and Department of Public Health, Fudan University, Shanghai, China; Professor Mathias Allemand, Department of Psychology, University of Zurich, Switzerland and Professor Sylvie Goldman, Columbia University, New York, USA.

### Dissemination plan

The study team plans to implement a knowledge translation using conventional methods for dissemination of scientific results, as well as modern communication strategies such as social media. The information about the study will be spread through established collaborations with non-profit organisations in Hong Kong and Shanghai, via Chinese American Psychoanalytic Alliance and the Shanghai Centre for Women's and Children's Health. We will use patient support groups, scientific and professional (midwifery, obstetric and psychological therapy), to communicate about our study.

We will use established scientific channels to communicate with the wider scientific community by publishing our protocol in an open access journal. We will also communicate our findings by presenting at national and international scientific meetings, along with publishing in a peer-reviewed international scientific journal.

We will also use more modern techniques, such as social media, to disseminate our findings to the public.

### DISCUSSION

This RCT will provide high-quality data about the effect of web-based psychosocial peer-to-peer intervention during pregnancy on maternal mental health. A recent review by Howard *et al*[55] has shown that psychological and psychosocial interventions could be effective treatments for postnatal depression. Some studies from low-income and middle-income countries have shown that these treatments can be provided effectively by trained peers or non-specialist workers.[56] Evidence on the treatment of antenatal depression is limited to small trials (with 36–53 women) of interpersonal therapy, culturally relevant brief interpersonal psychotherapy and CBT.[57–59] Little research exists about the effectiveness of such interventions (including web based) for other perinatal mental health problems.[55]

The information generated by this project could provide the basis for public health policies to be put in place and resource allocation for supporting pregnant and postpartum women suffering from depression, anxiety and PTSD, as a result of the COVID-19 pandemic.

The implementation of a culturally sensitive maternal mental health education addresses the need for more awareness of mental health issues in antenatal care. Our RCT can fill a knowledge gap and provide information on effectiveness of antenatal interventions provided by trained peers. Several studies have shown the effect of antenatal maternal mental health problems on child outcomes postpartum.[60–62] Social and emotional support reduces associations between postnatal depression and early cognitive development in the child.[63] The lack of recognition of perinatal mental health, distress and disorders may have serious implications for negative long-term effects on both mother and infant.[62 64] Therefore, it is important to recognise the transgenerational effect of perinatal mental health disorders and to intervene early to avoid negative outcomes for the child.

The study's assessment of women's mental health status at the end of the pregnancy provides information on the expected benefits of psychosocial support during antenatal care, with emphasis on cost-effective web-based psychosocial peer-to-peer support as an important alternative care model. Two recent studies have shown that as many as three out of four pregnant women use a pregnancy-related internet application and find these applications beneficial and trustworthy.[65 66] This is likely to be even more relevant in urban dwellers.

The routines in perinatal mental health screening policies share similarities, yet they are still different in some aspects in Hong Kong and Shanghai. In Shanghai, first universal prenatal health screening policies have been put in place recently, that is, at Shanghai Jiading District Maternity and Child Health Care Hospital and Obstetrics and Gynecology Hospital of Fudan University.[67 68] Currently, Shanghai provides only limited universal screening for perinatal mental health problems and lacks universal screening on provincial level. Adequate care for all pregnant women with increased risk for developing perinatal mental health problems is currently not available in Shanghai. In Hong Kong, however, universal PPD screening is one of the components of the Comprehensive Child Development Service in maternal and child health clinics, in collaboration with the Hospital Authority and Social Welfare Department.[69] Further, Hong Kong women attending perinatal healthcare can get access to a webpage provided by the government, promoting the spread of public health information as well as raised awarness about perinatal depression and services provided for pregnant and postpartum women and their families.[69] The resource allocation for universal screening and access to information might be crucial in order to reduce the burden of disease caused by perinatal mental health disorders on the family and the society. Our study compares the impact of intervention in two different healthcare settings.

As for most randomised trials, the external validity/generalisability of our findings could be limited since we will only target urban-dwelling population who are comfortable in using internet-based applications.

**Author affiliations**
[1]CLINTEC, Karolinska Institute, Stockholm, Sweden
[2]Women's Health and Perinatology Research Group, Department of Clinical Medicine, UiT-The Arctic University of Norway, Tromsø, Norway
[3]Psychiatry, The University of Hong Kong, Hong Kong, Hong Kong
[4]Psychological and Brain Sciences, Johns Hopkins University, Baltimore, Maryland, USA
[5]Counseling Psychology, National Taipei University of Education, Taipei, Taiwan

[6]Department of Research and Education, Tongji University, Shanghai, Shanghai, China
[7]Anthropology, The Chinese University Hong Kong, Hong Kong, Hong Kong
[8]Department of Women's and Children's Health, Karolinska Institutet, Stockholm, Sweden
[9]CLINTEC Department of Clinical Technology, Karolinska Institutet, Stockholm, Sweden

**Correction notice** This article has been corrected since it was published online. The license type has been updated from CC BY-NC to CC BY.

**Acknowledgements** The Mental health of Urban Mothers trial in urban Hong Kong and Shanghai is supported by Karolinska Institutet, Chinese University of Hong Kong, and Shanghai Government's Centre for Women's and Children's Health. We would also like to acknowledge the non-profit organisations for their support with patient recruitment.

**Contributors** SES wrote the study protocol article with the support of GA and EA. SES and GA conceived the study and were responsible for formulating initial study design together with H-YH local PI in Hong Kong responsible for conducting the study in Hong Kong, H-FC is responsible for recruitment of participants and set up all epidemiological background to imitate the study in Hong Kong and Shanghai. LZ is the local PI in Shanghai and together with LD responsible for running the study in Shanghai. MH was contributed to writing the protocol and conducted a literature search. S-CF is responsible for recruitment of participants. All authors were involved in development of study design and take responsibility for the final study design. SES wrote the first draft of the study protocol. All authors have contributed to the protocol development and have read and approved the final protocol and this manuscript.

**Funding** This project is funded by Swiss National Science Foundation SNSF (P2SKP3_187728).

**Competing interests** None declared.

**Patient and public involvement** Patients and/or the public were not involved in the design, or conduct, or reporting, or dissemination plans of this research.

**Patient consent for publication** Obtained.

**Provenance and peer review** Not commissioned; externally peer reviewed.

**Open access** This is an open access article distributed in accordance with the Creative Commons Attribution 4.0 Unported (CC BY 4.0) license, which permits others to copy, redistribute, remix, transform and build upon this work for any purpose, provided the original work is properly cited, a link to the licence is given, and indication of whether changes were made. See: https://creativecommons.org/licenses/by/4.0/.

**ORCID iD**
Simone Eliane Schwank http://orcid.org/0000-0002-1955-1123

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
