## [Reviewer comments · BMJ Open]

ARTICLE DETAILS

TITLE (PROVISIONAL)	Mental Health of Urban Mothers (MUM) Study: A multi-center randomized controlled trial, study protocol
AUTHORS	Schwank, Simone Eliane; Chung, Ho-Fung; Hsu, Mandy; Fu, Shih-Chien; Du, Li; Zhu, Liping; Huang, Hsuan-Ying; Andersson, Ewa; Acharya, G

VERSION 1 – REVIEW

REVIEWER	Claudia Ravaldi CiaoLapo Foundation, Italy
REVIEW RETURNED	19-Aug-2020

GENERAL COMMENTS	The Mental Health of Urban Mothers (MUM) Study is a well designed research aiming at addressing a relevant issue in a multicultural frame. Web-based psychosocial peer-to-peer support interventions could be of great interest for the prevention and the management of perinatal mental health disorders, also in order to reduce adverse pregnancy outcomes. The study is well conceived, the methodology is sound, the chosen tests seem adequate and the researchers appropriately qualified for its realization. I do not have any concerns or modifications to suggest.
--

REVIEWER	Leslie E. Roos University of Manitoba, Canada
REVIEW RETURNED	30-Aug-2020

GENERAL COMMENTS	Abstract - Could benefit from noting that data has already been published on the higher than expected rates of PPD during the pandemic Introduction The writing in the introduction would benefit from clarity and refinement in some areas: 1) The start of the introduction is somewhat confusing regarding the relevance of the trial to PPD, generally, or the COVID19 context, specifically. It would be helpful from the start to more clearly delineate if this was a planned trial that happened to occur within the pandemic or if it was specifically designed for this unique context. 2) The multisite and international nature of the trial is certainly a strength, but it would be helpful to from the introduction to discuss cross-national similarities in PPD risks and to later describe any theorized differences regarding PPD and trial relevance.
---

	3) Certain sentences are confusing (p3): With regards to risk perception, perceiving to be above the average total risk for woman herself or her infant increases the levels of uncertainty (23).  - More information regarding the evidence base to date of Thinking Healthy and other mHealth interventions would be very helpful. In general, the introduction would benefit from more structured organization regarding the evidence base of related research on the interventions and on maternal mental health during COVID. - The intervention focuses on PPD and anxiety, but the cited literature focuses on PTSD, which is confusing. Methods  - The outcome assessments would benefit from additional detail regarding reliability and validity of the measures to other gold-standard of psychodiagnostic interviews - The sample size and power calculation is based on a 20% reduction in scores, which is reasonable, but more information is needed on the clinical relevance of this score. - A consort diagram and additional information about to whom the program will be offered (i.e. is there universal screening at the clinics?) would strengthen the manuscript - The timeline of the expected EPDS and GAD7 scores to be compared across trial timepoints would benefit from more clarity. If data is collected continuously, which timepoints are to be used for the outcome calculations - Additional information on who will be included in the outcome statistics would be helpful – e.g. is this the full intent to treat sample? Based on actually engaging in peer support? Statistics  - I am not a statistician, but my understanding is that these clinical trial statistics would be more accurate if they included additional information regarding the expected correlations of measures across time points - It would be more appropriate to use a p-value of .025, given the use of 2 primary outcome measures
--	--

VERSION 1 – AUTHOR RESPONSE

Reviewer: 1

Reviewer Name: Claudia Ravaldi

Institution and Country: CiaoLapo Foundation, Italy

Please state any competing interests or state 'None declared': None declared

Please leave your comments for the authors below

The Mental Health of Urban Mothers (MUM) Study is a well designed research aiming at addressing a relevant issue in a multicultural frame.

Web-based psychosocial peer-to-peer support interventions could be of great interest for the prevention and the management of perinatal mental health disorders, also in order to reduce adverse pregnancy outcomes.

The study is well conceived, the methodology is sound, the chosen tests seem adequate and the

researchers appropriately qualified for its realization.
I do not have any concerns or modifications to suggest.

Author response:

We sincerely thank the reviewer for her positive comments.

Reviewer: 2

Reviewer Name: Leslie E. Roos

Institution and Country: University of Manitoba, Canada

Please state any competing interests or state 'None declared': None declared

Please leave your comments for the authors below

Abstract

- Could benefit from noting that data has already been published on the higher than expected rates of PPD during the pandemic

Author response: We have added this information in the abstract.

Introduction

The writing in the introduction would benefit from clarity and refinement in some areas:

1) The start of the introduction is somewhat confusing regarding the relevance of the trial to PPD, generally, or the COVID19 context, specifically. It would be helpful from the start to more clearly delineate if this was a planned trial that happened to occur within the pandemic or if it was specifically designed for this unique context.

Author response: Thank you for this comment. We have revised the introduction to give it a better structure and more clarity. This particular trial was designed considering the unique context of COVID-19 pandemic that started in China.

2) The multisite and international nature of the trial is certainly a strength, but it would be helpful to from the introduction to discuss cross-national similarities in PPD risks and to later describe any theorized differences regarding PPD and trial relevance.

Author response: We describe on p. 3 the cross-national and globally high and similar rates of PPD as well as the possibility of cross-national differences in risk factors and impact of pandemic on the prevalence of perinatal mental health disorders that is relevant to this trial.

3) Certain sentences are confusing (p3): With regards to risk perception, perceiving to be above the average total risk for woman herself or her infant increases the levels of uncertainty (23).

Author response: Thank you for the comment. We have revised the sentences including the one mentioned above.

- More information regarding the evidence base to date of Thinking Healthy and other mHealth interventions would be very helpful. In general, the introduction would benefit from more structured organization regarding the evidence base of related research on the interventions and on maternal mental health during COVID.

Author response: We have added further information on the Thinking Healthy intervention, and cited references that support the feasibility and evidence of its effectiveness.

- The intervention focuses on PPD and anxiety, but the cited literature focuses on PTSD, which is confusing.

Author response: We have mainly focused on PPD and anxiety and cited relevant literature in the revised manuscript.

Methods

- The outcome assessments would benefit from additional detail regarding reliability and validity of the

measures to other gold-standard of psychodiagnostic interviews

Author response: On P.7 We have given a rationale for the choice of scales over psychodiagnostic interviews and added relevant reference.

- The sample size and power calculation is based on a 20% reduction in scores, which is reasonable, but more information is needed on the clinical relevance of this score.

Author response: On P.7 We have provided further information regarding the clinical relevance and cited a references to support our statement.

- A consort diagram and additional information about to whom the program will be offered (i.e. is there universal screening at the clinics?) would strengthen the manuscript

Author response: We have provided information and discussed differences in antenatal screening programs in Shanghai and Hong Kong in discussion section. The intervention program will be offered to women randomized to the intervention arm of the trial after the recruitment.

- The timeline of the expected EPDS and GAD7 scores to be compared across trial timepoints would benefit from more clarity. If data is collected continuously, which timepoints are to be used for the outcome calculations.

Author response: Thank you for this comment. Under statistical methods, we have clarified that EPDS, GAD7 and IES-R scores will be compared between intervention and control groups and across time points (prenatal and postnatal) and the correlation between maternal antenatal mental health characteristics assessed by EPDS, GAD7 and IES-R at < 34-36 weeks and at 6 weeks postpartum.

- Additional information on who will be included in the outcome statistics would be helpful – e.g. is this the full intent to treat sample? Based on actually engaging in peer support?

Author response: Yes, it is the intention to provide peer-support two times during the antenatal period (i.e. at 18-22 weeks and at 24-28 weeks). This information is provided in the Figure 1.

Statistics

- I am not a statistician, but my understanding is that these clinical trial statistics would be more accurate if they included additional information regarding the expected correlations of measures across time points

Author response: We have added clarifying information on the statistical analysis.

- It would be more appropriate to use a p-value of .025, given the use of 2 primary outcome measures

Author response: Thank you for noticing this error. In a RCT it is best to have a single primary outcome measure and our primary outcome measure is EPDS as registered in the clinical trial registry. GAD7 is one of secondary outcome measures. The sample size was calculated based on EPDS only and the a two-sided p-value of 0.05 was chosen.

FORMATTING AMENDMENTS (if any)

Required amendments will be listed here; please include these changes in your revised version:

- Supplementary File

We have noticed that you have uploaded the file “Informed Consent MUM trial“ under 'supplementary file'. However, we cannot see any citation for this file within the main text. If this file needs to be published as supplementary file, please cite it as 'supplementary file' in the main text. Otherwise, kindly change its file designation to 'Supplementary file for editors only'.

Author response: We have changed it to 'Supplementary file for editors only'.

Thank you for your consideration. We look forward to hearing from you.

VERSION 2 – REVIEW

REVIEWER	Claudia Ravaldi CiaoLapo, Italy
REVIEW RETURNED	06-Oct-2020
GENERAL COMMENTS	I think that authors made all the corrections requested, so the manuscript is suitable for publication.